# Residential neighbourhood greenspace is associated with reduced risk of cardiovascular disease: A prospective cohort study

Alice M. Dalton[1,2], Andrew P. Jones[1,2]*

**1** Norwich Medical School, University of East Anglia, Norwich Research Park, Norwich, Norfolk, United Kingdom, **2** Centre for Diet & Activity Research (CEDAR), MRC Epidemiology Unit, University of Cambridge, Cambridge , United Kingdom

* a.p.jones@uea.ac.uk

## Abstract

### Background

Living in a greener neighbourhood may reduce the risk of developing incident cardiovascular disease, but evidence is limited by reliance on cross-sectional comparisons. We use data from a longitudinal study with a time-independent measure of risk to explore the association between exposure to greenspace and cardiovascular disease.

### Methods

Data was from the European Prospective Investigation of Cancer Norfolk UK cohort, baseline 1993–1997 (n = 24,420). Neighbourhoods were defined as 800m radius zones around participants' home, according to their home postcode (zip code) in the year 2000. Greenspace exposure was identified using classified satellite imagery. Adjusted Cox proportional hazards regression examined associations between greenspace and incident cardiovascular disease. Mediation analysis assessed if physical activity mediated associations, whilst modification by rurality, socio-economic status and age was explored.

### Results

The mean age of participants was 59.2 years at baseline, 54.7% were female, and mean follow-up time was 14.5 years. Individuals living in the greenest neighbourhood quartile had a 7% lower relative hazard of developing cardiovascular disease than other neighbourhoods (HR 0.93; 95% CI 0.88, 0.97; p = 0.003) after adjusting for age, sex, BMI, prevalent diabetes and socio-economic status (SES). Physical activity did not mediate the relationship (greenest compared to the least green quartile HR 0.99; 95% CI 0.97, 1.01; p = 0.416). Models predicted incidence of cardiovascular disease in the least green neighbourhoods (19.4% greenspace on average) would fall by 4.8% (95% CI 1.6%, 8.2% p = 0.003) if they were as green as the average neighbourhood (59.0% greenspace). Occupation moderated the relationship, whereby exposure to greenspace was not associated with incident CVD for participants engaged in manual occupations.

**Data Availability Statement:** Data from this study are available upon request to the EPIC-Norfolk Management Committee via their website http://www.epic-norfolk.org.uk, by email (epic@srl.cam.

ac.uk), or by phone 0800 616911. The EPIC-Norfolk study depends on data from NHS digital or its previous equivalent bodies. The data at individual record level are not allowed to be shared without having a data sharing agreement in place. Although the data are anonymised, the participants were recruited through general practices from a single region. The data sharing agreement ensures that collaborating researchers will not make contact with participants and their general practitioners.

**Funding:** This work was supported by the Centre for Diet and Activity Research (CEDAR), a UKCRC Public Health Research Centre of Excellence. Funding from Cancer Research UK, the British Heart Foundation, the Economic and Social Research Council, the Medical Research Council, the National Institute for Health Research, and the Wellcome Trust, under the auspices of the UK Clinical Research Collaboration, is gratefully acknowledged. The corresponding and lead authors were supported by MR/K025147/1. The views and opinions expressed herein are those of the authors. The funders had no role in study design, data analysis, the decision to publish, or the preparation of the manuscript.

**Competing interests:** The authors have declared that no competing interests exist.

## Conclusions

Greener home neighbourhoods may protect against risk of cardiovascular disease even after accounting for SES, whilst the mechanism does not appear to be strongly associated with physical activity. Putative causal mechanisms require investigation.

## 1. Introduction

Diseases of the heart and circulatory system (cardiovascular disease, CVD) are the biggest cause of death globally [1]. A total of 3.9 million die every year from CVD in Europe at a cost of €210 billion, whilst there were 11.3 million new cases of the disease in Europe in 2015 [2]. CVD accounts for 26% of all deaths in the UK [3]. It is estimated that two thirds of the burden of disease and over 14% of disability-adjusted life-years (DALYs) in the UK were due to poor diet and physical inactivity in 2010, both major risk factors for CVD [4]. Therefore, much of the burden of CVD is potentially preventable.

According to a substantial body of evidence [5–9], people have better physical and mental health in areas of greenspace–natural areas or urban vegetation [10]. A number of studies have specifically investigated the association between levels of greenspace and CVD [11–23]. Overall findings suggest a small yet statistically significant decrease in mortality for people living in greener areas, with meta-analyses suggesting a 0.96 reduction in risk of CVD (95% CI 0.94, 0.97) when comparing high verses low greenspace areas [8]. Of the two studies conducted in the UK, both looked at mortality from CVD. One found no significant association with greenspace exposure (proportion of land which was green in cities with a population of $\geq$100,000) [11], whilst another (exposure as proportion of land which was green, in census zones for urban areas with populations >10,000), found a statistically significant association only for men [17]. Some non-UK studies used self-report data for incident CVD, but this may be subject to error, particularly due to the occurrence of false positives for cerebrovascular accident [24, 25]. All existing CVD studies have examined prevalent CVD and have used cross-sectional data [26], and the vast majority employed mortality as the outcome. Prevalence data only allows the examination of relative risk; information on incidence is necessary to examine hazard ratios, which allow the examination of survival over time [27]. This is particularly pertinent for CVD development, as risk increases with age [28].

Potential causative routes explaining the association between greenspace and health have been explored [29, 30]. There may be general restorative [30] and salutogenic properties of living near or using greenspace, such as pleasure from viewing, relaxing, interacting with others, or just in knowing it is there [29]. It may also be greenspace promotes greater levels of physical activity [6, 31, 32], an aetiological factor in incident CVD [33–35]. Two notable studies have found physical activity to partly mediate the association between greenspace and cardiovascular health; one conducted in New Zealand (n = 8157) using self-report CVD data [18] and the other in Australia (n = 4041), using clinically-measured risk-markers for cardiometabolic disease [31]. Further research is required to establish if physical activity mediates the association between greenspace and clinically diagnosed incident CVD. If greenspace is beneficial for cardiovascular health via physical activity, such spaces may need to be designed in a way that they are accessible and suitable for facilitating this use.

This study uses data from a longitudinal prospective cohort study to investigate the association between neighbourhood greenspace and incident CVD of the residents over the course of follow-up, across a large population sample. Incident CVD was verified using hospital

admissions data, and greenspace was quantified using a detailed, objective measure. We also explored if physical activity was on the causal pathway between exposure to greenspace and incident CVD.

## 2. Methods

Information on the study design and setting, exposure to neighbourhood greenspace, physical activity and some of the covariates and confounders has been published previously, as a similar approach to the methods and analyses was used for this study [36]. This study advances previous work by examining a different health outcome, which is an important public health concern linked to modifiable lifestyle behaviours.

### 2.1 Study design and setting

Data was obtained from a prospective cohort study conducted in Norfolk, UK: the 'European Prospective Investigation of Cancer (EPIC) Norfolk' study [37, 38]. A total of 25,639 patients registered at 35 general practices were recruited from 1993 to 1997, and follow-up is still ongoing at the time of writing. During this period, self-reported and physician-obtained data on health and lifestyle was collected. For this analysis we use data from baseline and follow-up to 31 March 2016, up to when data for complete CVD ascertainment was available (Table 1).

Permission to take part in the study was gained from participants via informed consent. The Norwich District Ethics Committee granted ethical approval for the study.

### 2.2 CVD case ascertainment and verification

The primary endpoint was the date of incident CVD. Cohort participants were identified as having had incident CVD if they experienced any of the following between baseline and end of March 2016: ischaemic heart disease (angina, myocardial infarction and other ischaemic heart disease, International Statistical Classification of Diseases (ICD-10) [39] codes I20-I25) or cerebrovascular disease (haemorrhage, cerebral infarction, stroke, occlusion and stenosis of arteries, and other cerebrovascular diseases, ICD-10 codes I60-I69). Linkage to the National Health Service (NHS) England hospital information system and ENCORE (East Norfolk Commission Record) permitted the identification of hospital admission episodes and associated incident CVD from the hospital discharge code or death certificate. If a participant did not have incident CVD, their endpoint in the study was defined as 31 March 2016 or their date of death, if before then.

### 2.3 Exposure to neighbourhood greenspace

Methods used previously [36] were used to define the exposure variable; greenspace coverage (percentage) in the home neighbourhood of the participant. Briefly, the home postcode (zip

**Table 1. Variables used with type of measurement and date collected.**

| Variable | Measurement | Survey phase | Date collected |
|---|---|---|---|
| Incident CVD | Survey, GP records, hospital data | 18 month follow-up<br>3 years post-baseline<br>10 years post-baseline<br>19 years post-baseline | 1994–1998<br>1996–2000<br>2003–2007<br>2012–2016 |
| Home postcode (residential location) | From home address retained in administration records | N/A | 2000 |
| Physical activity | Survey (Health and Lifestyle Questionnaire) | Baseline | 1993–1997 |
| Height and weight | Physical examination by trained staff | Baseline | 1993–1997 |
| Demographics, lifestyle and health | Survey (Health and Lifestyle Questionnaire) | Baseline | 1993–1997 |

code) of each participant in the year 2000 was georeferenced using UK Ordnance Survey methodology [40]. Using geographic information system (GIS) software [41], neighbourhoods were defined as 800m area boundaries around the home postcode, using straight line distance to create a circular buffer around the home. The postcode represents the most detailed level of resolution available for home neighbourhood analysis, as each of the 1.3 million postcodes in England and Wales covers on average only 42.8 people (min 1, max 3215; st dev 38.8) and 17.9 households (min 0, max 646; st dev 14.8) [42]. Each neighbourhood was overlaid with a digital land cover map [43] and the amount (percentage) of the area that was classed as greenspace was calculated. Greenspace was defined as woodland, arable land, grassland, mountain, heath or bog. As different neighbourhood boundaries can be chosen for analysis of environmental exposure [44–46], we compared results using three different distances from the home postcode (800m, 3km and 5km). For further sensitivity analysis, we also measured neighbourhoods as the area around the home that could be accessed by travelling on the road network (road buffer) rather than by straight line distance.

## 2.4 Physical activity

Previous studies using the EPIC-Norfolk cohort have found that physical activity is associated with a lower likelihood of developing CVD [47–49]. Therefore we undertook analysis to see if greenspace was associated with lower likelihood of incident CVD in the cohort through the mediating role physical activity. Physical activity was quantified based on participants' responses to two questions in the baseline Health and Lifestyle questionnaire: the type and amount of physical activity involved in their work (four options of sedentary, standing, physical and heavy manual), and the amount of recreational activity undertaken in a typical week in the last 12 months, measured according to two different categories of cycling and other physical exercise (keep fit, aerobics, swimming and jogging) and reported separately for summer and winter [48]. Using this information, the study coordinators had created a categorical measure of physical activity [50], whereby participants were assigned to a category of active, moderately active, moderately inactive or inactive. The cut-off points are detailed in Table 2.

## 2.5 Covariates and confounders

Health and lifestyle characteristics were collected in the baseline Health and Lifestyle Questionnaire. Variables used for the analysis, selected for their potential role in the association between greenspace and incident CVD, included age, sex, BMI (from measured height and weight data), self-reported prevalent diabetes, individual socio-economic status (SES) [44] (measured by occupation [51]) and neighbourhood socio-economic status (Townsend Index area deprivation [52]). We could not explore the potential link between ethnicity and CVD [53], as 99.7% of respondents were white British.

**Table 2. Categories of physical activity with description.**

| Category | Description |
|---|---|
| Inactive | Sedentary job and no recreational activity |
| Moderately inactive | Sedentary job with <0.5 hrs recreational activity per day; or standing job with no recreational activity |
| Moderately active | Sedentary job with 0.5–1 hour recreational activity per day; or standing job with 0.5 hrs recreational activity per day; or physical job with no recreational activity |
| Active | Sedentary job with >1 hrs recreational activity per day; or standing job with >1 hrs recreational activity per day; or physical job with at least some recreational activity; or heavy manual job |

## 2.6 Data analysis

A total of 14 of the participants did not have a valid postcode that could identify their residential location in the year 2000, so were excluded from the full sample of 25,639 at baseline. Participants living more than 20 km from Norfolk in 2000, and were hence outside the margins of the greenspace database, were also not included in the analysis (n = 1189). Thirteen participants who already had incident CVD between when they were first recruited to the study and the baseline survey were not included in the analysis. Three further participants opted out of the study after baseline so were also excluded, leaving a final sample of 24,420.

Cox proportional hazards regression modelling was performed [54] to estimate the probability of each participant experiencing incident CVD, according to their exposure to greenspace. Hazard ratios were calculated using length of follow-up before disease incident. Time in study was defined as participant age, commencing with their age at baseline. The end point was defined as the participant's age at first of the following; incident CVD event (hospital admission or death), death from other causes, when they were lost to follow-up or withdrew from the study, or at end of follow-up (31 March 2016). If participants were lost to follow-up, CVD outcome data could still be accessed using hospital records.

Two Cox models were used. Model 1 included only the primary exposure, neighbourhood greenspace, which was split into quartiles. Model 2 was a multivariate model, which additionally adjusted for potential confounders of age, sex, BMI, SES and baseline (prevalent) diabetes. The association between exposure and outcome was illustrated using Kaplan-Meier plots [55]. In addition, the population attributable fraction (PAF) was computed to estimate the extent to which incident CVD could be reduced if participants living in the least green areas were all exposed to greener environments (based on the average percentage of greenspace observed in the cohort). This used the command 'punafcc' in Stata [56, 57].

The proportional hazards assumption was tested to see if the hazard ratios were constant over time with Schoenfeld residuals, piecewise analysis (time divided into quartiles) to identify time-varying covariates, and adding interaction terms between covariates and time to the model to see if hazard ratios varied over time [58]. In addition, models were tested for interaction effects according to variables selected a priori according to evidence in the literature and assessed model fit using the likelihood ratio test. We tested the models for interactions between greenspace and urban-rural status, to investigate if the urban or rural nature of greenspace potentially changes the association with incident CVD. In addition, as greenspace use has been shown to be spatially patterned according to SES [59, 60], we stratified models according to individual SES (manual versus non-manual occupation), to see if this had a modifying effect on the data. We also stratified by age to see if it modifies the association between greenspace and CVD, as research suggests that greenspace use differs for older versus younger older adults [61].

Mediation analysis was carried out to assess if physical activity explained any associations between greenspace and incident CVD. Commonly-used mediation testing [62–64] is not suitable for use with survival data, as they often violate assumptions of normality and are right-censored [65]. Instead, procedures set out by Lange et al were used as they are well-suited to use with survival analysis [66, 67]. The direct and indirect effects between exposure and outcome were estimated and reported as hazard ratios. The statistical software package R was used for the mediation analysis [68], and Stata v13 was used for the remaining analyses [69].

# 3. Results

## 3.1 Sample characteristics

In total, 4765 participants were known to have moved to a new house between 2000 and 2014 (19.5%). 1484 participants provided just one postcode during the study, so we do not know if

**Table 3. Baseline characteristics of included participants by gender.**

| Characteristic | Men (n = 11067) | Women (n = 13353) | All (n = 24420) |
|---|---|---|---|
| Age (years) | 59.6 ± 9.3 | 58.8 ± 9.3 | 59.2 ± 9.3 |
| BMI (kg/m$^2$) | 26.5 ± 3.3 | 26.2 ± 4.4 | 26.4 ± 3.9 |
| Duration of follow-up (years) | 13.6 ± 6.4 | 15.2 ± 5.9 | 14.5 ± 6.2 |
| Social class (%) | | | |
| Professional | 7.6 (831) | 6.5 (842) | 7.0 (1673) |
| Managerial | 38.1 (4143) | 35.2 (4576) | 36.5 (8719) |
| Skilled non manual | 12.6 (1373) | 19.9 (2585) | 16.6 (3958) |
| Skilled manual | 25.4 (2758) | 21.2 (2761) | 23.1 (5519) |
| Semi-skilled | 13.3 (1448) | 13.4 (1739) | 13.3 (3187) |
| Unskilled | 2.9 (318) | 3.9 (514) | 3.5 (832) |
| Townsend Index of deprivation[a] (index) | -2.1 ± 2.2 (11042) | -2.0 ± 2.2 (13313) | -2.0 ± 2.2 (24355) |
| Prevalent diabetes (%) | 3.1 (344) | 1.6 (216) | 2.3 (560) |
| Overall physical activity (%) | | | |
| Inactive | 31.1 (3442) | 30.5 (4071) | 30.8 (7513) |
| Moderately inactive | 24.5 (2708) | 32.0 (4277) | 28.6 (6985) |
| Moderately active | 22.8 (2527) | 22.2 (2959) | 22.5 (5486) |
| Active | 21.6 (2389) | 15.3 (2046) | 18.2 (4435) |
| Leisure physical activity (hrs per wk cycling/sport, % per category) | | | |
| 0 | 55.4 (6136) | 51.3 (6827) | 53.3 (12963) |
| >0-<3.5 | 26.4 (2926) | 32.8 (4394) | 29.9 (7320) |
| > = 3.5-<7 | 11.1 (1232) | 10.5 (1405) | 10.7 (2637) |
| > = 7 | 7.0 (773) | 5.4 (727) | 6.1 (1500) |
| Walking in summer (hrs per wk) | 10.1 ± 11.0 | 9.4 ± 10.1 | 9.7 ± 10.5 |
| Urban/rural location (%) | | | |
| Urban | 63.6 (6607) | 64.2 (8014) | 63.9 (14621) |
| Town and fringe | 10.7 (1113) | 10.7 (1334) | 10.7 (2447) |
| Rural | 25.7 (2676) | 25.1 (3139) | 25.4 (5815) |
| Greenspace (% < = 800m home) | 59.3 ± 29.8 | 58.9 ± 29.9 | 59.0 ± 29.8 |

Values are % (n) or mean ± SD.

[a] A standardised index of between −6.7 (relatively affluent) to +7.0 (relatively deprived), where a score of 0 represents an area with overall mean values.

they moved during that period. There were small but statistically significant differences between those excluded and those included in the analysis (final sample, n = 24,420), for age (mean 60.6 years in the excluded, 59.2 years in the included, p = <0.001), incident CVD (58.0% in the excluded, 54.4% in the included, p = 0.014), rural home location (36.3% in the excluded, 30.7% in the included, p<0.001), and physical activity in terms of not taking part in any cycling or sports 57.5% in excluded, 53.1% in the included, p = 0.012).

The baseline characteristics of included participants are presented in Table 3. A total of 13,279 (54.4%) participants experienced incident CVD between baseline and the end of March 2016, of which 5734 cases were fatal (43.2%). 1829 participants from the cohort died of other causes during follow-up (7.5%). A total of 15,727 (69.0%) participants were alive at the end of the observation period. The mean duration of follow-up observed across the cohort was 14.5 years.

### 3.2 Greenspace and incident CVD

Annual rates of incident CVD increased from 29.2 per 10,000 person-years (1 in 342) for those aged 40–49, to 1198.4 per 10,000 person years (1 in 8.3) for those aged 80–90 years. The

cumulative survival for the cohort not developing incident CVD was 36.9% at age 80, which decreased to 9.4% at age 90 (Fig 1A; data is only presented up to age 90 due to few observations after this).

There was a significant trend in the probability of remaining free from CVD across the quartiles of greenspace (p = <0.001) (Fig 1B); the probability of remaining free from CVD amongst those in the greenest areas was 4.5% higher than those in the least green at age 80, a figure that dropped to 0.9% at age 90.

Hazard ratios (HR), confidence intervals (CI) and p-values from the Cox models are presented in Table 4. Before adjustment for confounders, individuals living in the greenest quartile (Quartile 4) had a 8% lower relative hazard of developing incident CVD (HR 0.92; 95% CI 0.88, 0.97; p = 0.001) compared to those living in the least green quartile (Model 1). The linear trend across quartiles was statistically significant (HR 0.97; 95% CI 0.96, 0.99; p = <0.001). The hazard ratio was very slightly modified (HR 0.93; 95% CI 0.88, 0.97; p = 0.003) after adjusting for age, sex, BMI, prevalent diabetes and SES (Model 2), with a statistically significant trend across quartiles (HR 0.97; 95% CI 0.96, 0.99; p = 0.001).

Schoenfeld residuals suggested no evidence of violation of the proportional hazards assumption for the exposure to greenspace measure. However, the global test suggested a violation overall (p<0.001), with four individual covariates potentially violating the assumption (age, baseline diabetes, sex and BMI). Piecewise analysis investigating differences in hazard ratios according to different time points of the study (measured by quartiles of age to CVD incident) suggested that direction of association for age at baseline changed for people who experienced incident CVD in the later years of their life, whereby an older baseline age (Quartile 4, >79.7 years) reduced the hazard for incident CVD (HR 0.99) when compared with people experiencing CVD in younger years (Quartile 1, <68.1 years, HR 1.19). The findings for sex, baseline diabetes and BMI showed some variation over time, but with no clear trend. In order to test for the impact of any violation of the proportional hazards assumption, interaction terms between all four variables and time (age at CVD incident) were added in to the fully adjusted model, three of which were statistically significant (age p<0.001, baseline diabetes p = 0.011, and BMI p = 0.046), and which overall contributed to a statistically significant improvement in model fit (by p<0.001), but had little effect on exposure effect or statistical significance.

The PAF suggested that, based on the model before adjustment, incident CVD in the least green neighbourhoods (Quartile 1) would fall by 5.1% (95% CI 2.1%, 8.2%; p = 0.001) if those neighbourhoods were as green as the average neighbourhood across the whole sample (59% greenspace). After full adjustment, the corresponding estimate decreased slightly to 4.8% (95% CI 1.6%, 8.2% p = 0.003).

### 3.3 Assessment of mediation and moderation

Mediation analysis suggested that physical activity did not mediate the association between exposure to greenspace in the home neighbourhood and incident CVD (Table 5). A HR of 0.99 was attributed to the pathway through physical activity, for the greenest compared to the least green quartile, but this was not statistically significant (95% CI 0.97, 1.01; p = 0.437). When the covariates of age, BMI, sex, prevalent diabetes and SES (individual and neighbourhood) were accounted for, the HR remained non-statistically significant (HR 0.99; 95% CI 0.97, 1.01; p = 0.416).

As we were unable to incorporate change in exposure into the model with one single measure of greenspace, based on participants' home postcodes for the year 2000, sensitivity analysis was conducted whereby only participants who had not moved after 2000 were included in

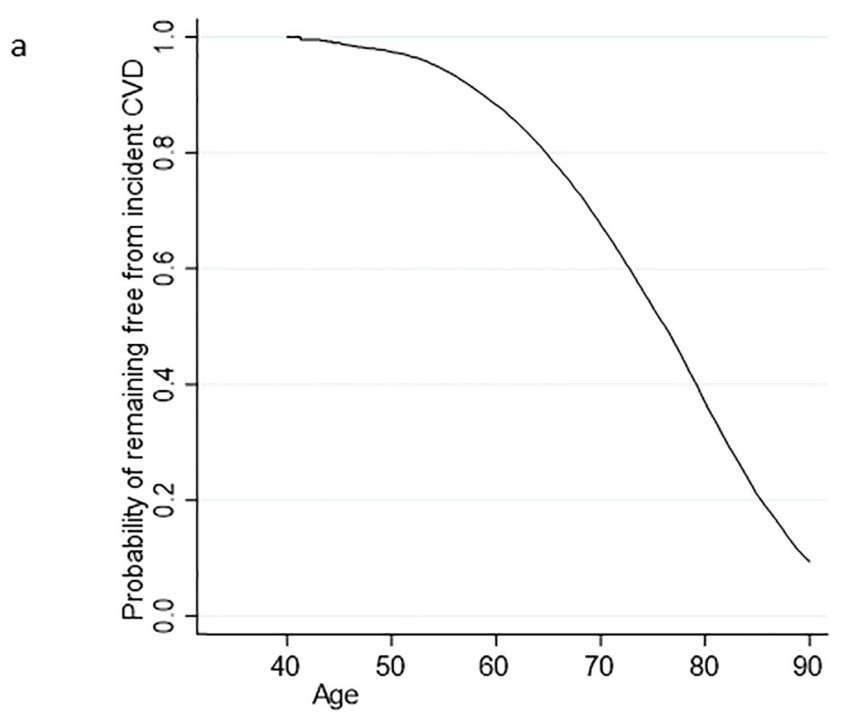

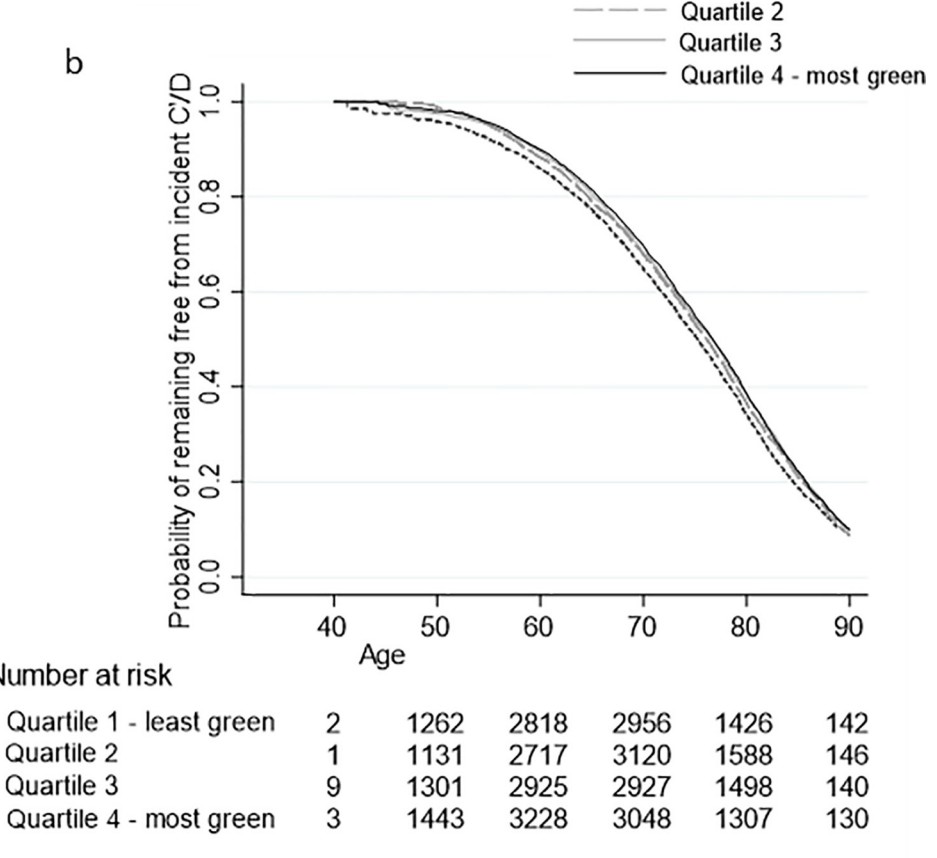

| Number at risk | | | | | | |
| --- | --- | --- | --- | --- | --- | --- |
| Quartile 1 - least green | 2 | 1262 | 2818 | 2956 | 1426 | 142 |
| Quartile 2 | 1 | 1131 | 2717 | 3120 | 1588 | 146 |
| Quartile 3 | 9 | 1301 | 2925 | 2927 | 1498 | 140 |
| Quartile 4 - most green | 3 | 1443 | 3228 | 3048 | 1307 | 130 |

**Fig 1. Kaplan-Meier survival curves showing the probability of remaining free of CVD since baseline using age as the underlying timescale.** (a) in the overall sample and (b) in categories based on quartiles of proportion of total land area of participants' home neighbourhood that is greenspace. Probabilities are only presented up to age 90 due to small numbers of participants older than this.

the model. Strength of effect, direction of association and statistical significance were largely unaffected (adjusted for age, sex, BMI, prevalent diabetes and SES, greenest versus least green quartile HR 0.92; 95% CI 0.86, 0.97; p = 0.004; n = 17,682).

There was a statistically significant interaction between occupation and greenspace in the regression models, where those living in greener areas and engaged in non-manual occupations had a lower hazard of incident CVD than those living in less green areas and engaged in manual occupations (trend across quartiles HR 0.97, p = 0.038). Adding the interaction terms to the model significantly improved model fit (likelihood-ratio test p = 0.026), suggesting that individual occupation modified the relationship between exposure to greenspace and incident CVD. In stratified analysis, exposure to greenspace was associated with incident CVD for participants engaged in non-manual occupations, with a dose-response relationship of lower hazard ratios with increasing level of greenspace (trend p<0.001, Table 6). The relationship was not statistically significant for those working in manual occupations (trend p = 0.829). There was no statistically significant interaction between greenspace and rurality (trend across greenspace quartiles p = 0.574), age (trend p = 0.529) or sex (p = 0.737) in the regression models.

Sensitivity analysis regarding characterising exposure to greenspace in the home neighbourhood found that greenspace measured using larger buffer sizes and road network buffers, showed similar statistically significant associations with incident CVD, both in terms of direct relationship and via the mediator of physical activity (S1 Table: Sensitivity testing).

## 4. Discussion

The findings of this study suggest that exposure to greenspace may be protective against incident CVD in older adults. Participants living in the greenest locations had a 7% lower relative risk of developing CVD at follow-up when compared to those living in the least green areas, after adjustment for potential confounding by age, sex, BMI, prevalent diabetes and SES. Overall physical activity, measured according to a four level classification based on occupation and leisure activity, was not found to explain any of the association between exposure to greenspace and incident CVD in adjusted analysis.

Interestingly, an association between greenspace and incident CVD was only apparent for those in non-manual jobs, suggesting that exposure to greenspace may be only protective

**Table 4. Hazard ratios from Cox regression, showing the association between neighbourhood greenspace exposure and incident CVD.**

| | | Model 1 Adjusted for greenspace (N = 22420) | | | | | | | Model 2 Adjusted for confounders (N = 23759) | | | | | |
| | | | | 95% CI | | | | | | | 95% CI | | | |
| | n | Person years (1000) | HR | Lower | Upper | p | p trend | n | Person years (1000) | HR | Lower | Upper | p | p trend |
|---|---|---|---|---|---|---|---|---|---|---|---|---|---|---|
| Greenspace quartile | | | | | | | <0.001 | | | | | | | 0.001 |
| 1 (least green, ref) | 6107 | 86.3 | 1.00 | | | | | 5935 | 84.1 | 1.00 | | | | |
| 2 | 6107 | 87.2 | 0.96 | 0.92 | 1.01 | 0.136 | | 5945 | 85.2 | 0.97 | 0.92 | 1.02 | 0.189 | |
| 3 | 6103 | 88.9 | 0.92 | 0.88 | 0.96 | 0.001 | | 5966 | 87.3 | 0.92 | 0.88 | 0.97 | 0.002 | |
| 4 (most green) | 6103 | 91.6 | 0.92 | 0.88 | 0.97 | 0.001 | | 5913 | 89.0 | 0.93 | 0.88 | 0.97 | 0.003 | |

Age is used as the underlying time scale. Model 2 adjusted for confounders of sex, age, BMI, prevalent diabetes and SES (individual and neighbourhood). CI: confidence interval.

**Table 5. Total, direct and indirect (via physical activity) effects of neighbourhood greenspace exposure and incident CVD.**

| | Model 1 Adjusted for greenspace (N = 24420) | | | | Model 2 Adjusted for confounders (N = 23759) | | | |
|---|---|---|---|---|---|---|---|---|
| | | 95% CI | | | | 95% CI | | |
| | HR | Lower | Upper | p | HR | Lower | Upper | p |
| Effect (most versus least green quartile) | | | | | | | | |
| Total effect | 0.92 | 0.88 | 0.97 | 0.001 | 0.93 | 0.88 | 0.97 | 0.003 |
| Direct effect | 0.93 | 0.91 | 0.96 | <0.001 | 0.94 | 0.91 | 0.96 | <0.001 |
| Indirect effect (through physical activity) | 0.99 | 0.97 | 1.01 | 0.437 | 0.99 | 0.97 | 1.01 | 0.416 |

Age is used as the underlying time scale. Model 2 adjusted for confounders of sex, age, BMI, prevalent diabetes and SES (individual and neighbourhood). CI: confidence interval.

against CVD for certain demographic groups. As non-manual occupations are more likely to work in indoor rather than outdoor settings, living near greenspace may matter more to them.

The findings are partially in agreement with two cross-sectional studies conducted in Australasia, which found that CVD disease risk was lower in greener neighbourhoods [18, 31]. On the other hand, both those studies found physical activity to be a statistically significant mediator in the association between greenspace and cardiovascular health, which although the effect sizes were small, contrasts our findings. Our findings agree with other studies looking at general health, which have found that overall physical activity does not explain why people living in greener neighbourhoods have poorer health [70]. Ours is the only study to use actual incident CVD, confirmed through hospital records, which was experienced by 13,279 individuals in the cohort (54.4% of the sample), 56.8% of whom were still alive at the end of follow-up. Richardson [18] used self-reported CVD incident data from 976 individuals (12% of the sample), and Paquet [31] estimated a risk score for 3574 individuals.

The causes of the association between greenspace exposure and incident CVD remain largely unexplained, despite a growing body of work in this field [30,71]. Other explanatory mechanisms are likely to explain the causal link such as the physiological and psychological benefits of seeing greenspace [72], including stress reduction and restoration [73]; the role of greenspace in creating a sense of attachment to place and community, and reduced isolation [74] and increased social ties [73]; the advantages of exposure to nature for immunological regulation [73,75–79]; and, albeit with less evidence, the 'biogenics' hypothesis, that natural toxins and compounds can reduce unhealthy cell activity in humans, and reduce the incidence of disease [73].

**Table 6. Hazard ratios from Cox regression, showing the association between neighbourhood greenspace exposure and incident CVD, stratified according to occupation.**

| | Model 3 Manual occupation (N = 9489) | | | | | | | Model 4 Non-manual occupation (N = 14270) | | | | | | |
|---|---|---|---|---|---|---|---|---|---|---|---|---|---|---|
| | | | | 95% CI | | | | | | | 95% CI | | | |
| | n | Person years (1000) | HR | Lower | Upper | p | p trend | n | Person years (1000) | HR | Lower | Upper | p | p trend |
| Greenspace quartile | | | | | | | 0.829 | | | | | | | <0.001 |
| 1 (least green, ref) | 2593 | 35.9 | 1.00 | | | | | 3342 | 48.3 | 1.00 | | | | |
| 2 | 2593 | 36.6 | 0.94 | 0.88 | 1.01 | 0.093 | | 3352 | 48.6 | 0.99 | 0.93 | 1.06 | 0.788 | |
| 3 | 2205 | 31.1 | 0.96 | 0.88 | 1.03 | 0.255 | | 3761 | 56.2 | 0.90 | 0.84 | 0.96 | 0.003 | |
| 4 (most green) | 2098 | 30.3 | 0.99 | 0.91 | 1.07 | 0.752 | | 3815 | 58.7 | 0.89 | 0.83 | 0.95 | 0.001 | |

Age is used as the underlying time scale. All models adjusted for confounders of sex, BMI, prevalent diabetes and neighbourhood SES. CI: confidence interval.

The prevalence of CVD in the EPIC-Norfolk cohort at 54.7% is higher than the prevalence of CVD in the British population, which is around 10% across all ages, and around 30% in people aged over 75 years [28]. EPIC-Norfolk is an elderly cohort, with an average age of 72.7 years for those still alive at the end of the follow-up (37.2% of those still alive (5844) were 75 or older) which offers some explanation. In addition, CVD events in the EPIC-Norfolk cohort are defined according to a very broad category of CVD. This includes some cerebrovascular diseases that are not always included in narrower definitions (ICD-10 codes I64-9).

The strengths of this research include the robust nature of the classification and dating of incident CVD, through linkage with hospital records along with self-reported incidents. The cohort provided a large sample size and length of follow-up, with 13,279 incident cases in older adults, to estimate associations between greenspace and incident CVD. In addition, we used a robust method of mediation appropriate for use with survival data, non-linear models and categorical mediators [80]. It is also noted that despite to the historical nature of the postcodes used for this analysis, only 0.06% of participant postcodes could not be matched with a geographical location (14 out of 24,434), representing a very small proportion of the total sample. Finally, we tested different classifications of exposure to greenspace by running the models on different neighbourhood buffer sizes and types (circular and network-based).

Some limitations of this research must be mentioned. For those that changed address during the study, we did not have data on the date or number of moves that took place. We therefore assumed their greenspace exposure did not vary during follow-up, and the sensitivity analysis suggested the results were not affected when movers were excluded from the study. Further, we do not know how long participants had resided at their address prior to the year 2000. Use of greenspace amongst our participants was not known, we do not know if residential locations represent activity spaces of individuals, and we did not use a measure of publicly accessible greenspace. However, it may be that just the presence of any greenspace may benefit health rather than actual use. We had no information on greenspace quality. The high-resolution land cover data we used was indicative of an area that could be accessed/viewed where people live, and the sensitivity of the measure was examined by varying the size and type of this neighbourhood area. We measured physical activity using self-reported data which may introduce error, although the measures used have been shown to be useful when compared with device-measured data in validation studies [50, 81]. Housing costs may be a potential confounding variable linked with deprivation, but we do not believe this, which is a relatively poorly measured variable in the UK, would add strong predicative power to the model over and above the impact of the Townsend Index.

Residual confounding from other factors, not considered as part of this analysis, may explain the association between greenspace and CVD risk, for example, via air pollution [16]. Our study area of Norfolk may not be generalisable to areas with substantially different environments or demographic structures. The sample was not ethnically diverse, as nearly all (99.7%) of people in the study sample were white. However, the cohort was distributed across a range of urban and rural settings in Norfolk, with a high diversity of greenspace exposure. Finally, we did not have data on behavioural factors, such as smoking, alcohol intake and diet, which are likely to be associated with neighbourhoods and communities.

## 5. Conclusions

There is evidence that people living in greener neighbourhoods are less likely to experience incident CVD, even after adjusting for neighbourhood SES. This association does not seem be explained by higher levels of overall physical activity. Until we understand the mechanisms through which cardiovascular health benefits occur, we cannot fully know how to design

effective interventions. It may be that improving aesthetics and view-points is the best method, to enable people to see greenspace. On the other hand, it may be necessary to improve access to places rich in species diversity to strengthen immune systems, or create opportunities for social interaction to reduce social isolation. Further research is needed to examine these potential causal mechanisms.

## Supporting information

**S1 Table. Sensitivity testing.**
(DOCX)

## Acknowledgments

We thank all EPIC-Norfolk participants and staff for their contribution to the study. We thank the following people for their assistance with this research: Simon Griffin, Robert Luben, Stephen Sharp and Nicholas J Wareham.

## Author Contributions

**Conceptualization:** Alice M. Dalton, Andrew P. Jones.

**Data curation:** Alice M. Dalton.

**Formal analysis:** Alice M. Dalton.

**Funding acquisition:** Andrew P. Jones.

**Investigation:** Alice M. Dalton, Andrew P. Jones.

**Methodology:** Alice M. Dalton.

**Software:** Alice M. Dalton.

**Supervision:** Andrew P. Jones.

**Validation:** Alice M. Dalton.

**Visualization:** Alice M. Dalton.

**Writing – original draft:** Alice M. Dalton.

**Writing – review & editing:** Alice M. Dalton, Andrew P. Jones.

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
