## [Decision Letter · Decision Letter 0]

30 Sep 2019

PONE-D-19-21826

Residential neighbourhood greenspace is associated with reduced risk of cardiovascular disease: a prospective cohort study

PLOS ONE

Dear Dr Jones,

Thank you for submitting your manuscript to PLOS ONE. After careful consideration, we feel that it has merit but does not fully meet PLOS ONE’s publication criteria as it currently stands. Therefore, we invite you to submit a revised version of the manuscript that addresses the points raised during the review process.

We would appreciate receiving your revised manuscript by Nov 14 2019 11:59PM. To enhance the reproducibility of your results, we recommend that if applicable you deposit your laboratory protocols in protocols.io, where a protocol can be assigned its own identifier (DOI) such that it can be cited independently in the future. For instructions see: http://journals.plos.org/plosone/s/submission-guidelines#loc-laboratory-protocols

We look forward to receiving your revised manuscript.

Kind regards,

David Meyre

Academic Editor

PLOS ONE

Journal Requirements:

2. Thank you for submitting the above manuscript to PLOS ONE. During our internal evaluation of the manuscript, we still found significant text overlap between your submission and the following previously published works, on which you are an author.

https://doi.org/10.1186/s12889-016-3833-z

Please revise the manuscript to rephrase the duplicated text, cite your sources, and provide details as to how the current manuscript advances on previous work. Please note that further consideration is dependent on the submission of a manuscript that addresses these concerns about the overlap in text with published work.

4. Thank you for stating the following financial disclosure: "The funders had no role in study design, data collection and analysis, decision to publish, or preparation of the manuscript."

Please provide an amended Funding Statement that declares *all* the funding or sources of support received during this specific study (whether external or internal to your organization) as detailed online in our guide for authors at http://journals.plos.org/plosone/s/submit-now.  Please state what role the funders took in the study.  If any authors received a salary from any of your funders, please state which authors and which funder. If the funders had no role, please state: "The funders had no role in study design, data collection and analysis, decision to publish, or preparation of the manuscript."

Reviewers' comments:

Reviewer's Responses to Questions

**Comments to the Author**

1. Is the manuscript technically sound, and do the data support the conclusions?

Reviewer #1: Partly

Reviewer #2: Partly

2. Has the statistical analysis been performed appropriately and rigorously? 

Reviewer #1: No

Reviewer #2: No

3. Have the authors made all data underlying the findings in their manuscript fully available?

Reviewer #1: No

Reviewer #2: No

4. Is the manuscript presented in an intelligible fashion and written in standard English?

Reviewer #1: No

Reviewer #2: Yes

5. Review Comments to the Author

Reviewer #1: Thank you very much for giving me an opportunity to read this interesting manuscript. This study used information collected in Norfolk, UK and investigated the prospective association between residential neighborhood greenness and the risk of developing cardiovascular diseases. I have several questions and suggestions for the authors.

[Abstract]

- Please specify which year baseline information was collected (1993-1997?).

- (L.27) The authors stated the mean follow-up period was 17.8 years, which contradicts with the numbers I calculated from Table 2 (14-15 years).

- (L.30) The sentence “Physical activity did not mediate the relationship (HR = 0.99, 95%CI 0.97, 1.02; p = 0.475)” is not clear. What does this point estimate mean?

[Introduction]

- (L.49-65) The second paragraph is redundant. Needs to be revised.

- (L.57) What does “self-reported prevalent CVD”? Citation #10 used information collected in the UK Small Area Health Statistics Unit, which were all registered deaths.

[Methods]

- Section numbers needs to be revised (2.1 → 2.1 → 2.2 → 2.1 → 2.2 → 2.3).

- (L.120-123) I disagree with their assumption. They only accounted for the period between 2000 and 2014 - but the participants had lived at the place for a certain amount of time already in 2000. In addition, for those who developed CVD and moved to another location, calculating the “equal time” for 2000 address and 2014 address means nothing. I recommend the authors just use baseline address for main analysis and conduct a sensitivity analysis in which they exclude those who moved after 2000. Their assumption is theoretically and statistically irrelevant.

- (L.126) Please provide more information on how PA was estimated (e.g., information on the questionnaire used, cut-offs to create categories...).

- (L.153) How did you define the last date of observation period for those who were lost to follow-up?

[Results]

- (L.180-185) These sentences should appear in Method section.

- (L.193) Please information on the numbers of [1] fatal CVD cases, [2] non-fatal cases, [3] those who died due to causes other than CVD, [4] those who were lost to follow-up (censored cases) and [5] those who were alive at the end of observation period.

- (L.196) Please check if the follow-up period is correct (17.8 years).

- (L.198) Why presented three types of PA results? I also don’t understand why they did not provide detailed information (hrs per week) for the last PA result.

- (L.236) Please clarify that models were adjusted for both individual- and neighborhood-level SES.

- Please provide results of proportional hazard assumption (e.g., Schoenfeld Residuals Test)

- (L.242) Please provide a table for mediation analysis.

[Discussion]

- (L.288-290) The biogenics hypothesis does not seem to be accepted as much as the other interpretations. I recommend the authors should delete (or at least tone down) the sentence.

- (L.308-309) Please try the sensitivity analysis mentioned above.

- Lack of information on the access to gym or swimming pool can be a limitation of this study. These places have nothing to do with greenness but provide an opportunity for the residents to engage in physical activities.

- I think the authors should emphasize that they observed the association between neighborhood greenness and incident CVD even after adjusting for neighborhood SES. Some might suspect that rich neighborhoods have a better environment (e.g., more parks) and that greenspace is just an confounder in the association between neighborhood SES and CVD, particularly because the authors did not show any evidence of mediation by physical activity.

Reviewer #2: This study adds to the building literature on greenspace and cardiovascular disease by modeling time-dependent outcomes in a population-based cohort of Britons from the county of Norfolk. Another interesting component of this manuscript is its analyses of effect mediation & moderation, establishing that most of the beneficial outcomes associated with nature exposure do not seem to be attributable to physical activity, and that there are interesting relationships among specific subpopulations defined according to occupation.

How granular are postcodes in the UK? In the US, researchers commonly consider census block group (600-3000 population) to be a suitable level of geographic resolution for characterizing neighborhood effects. Zip-code level characterizations are far more crude and can smooth over important geographic differences in exposures. Generally speaking, the result of using too large of areas for spatial epidemiologic studies is that it tends to bias results toward the null. The sensitivity analysis of different definitions of neighborhood exposure seem to address this issue, although it probably should be stated how many residents tend to live within a given postcode in the UK and additionally the description of the buffer-based neighborhood definitions needs to be made more specific (e.g., were these buffers defined in reference to individuals addresses of residence? What is a "road" buffer type?)

Cardiovascular risk is complex and is affected by physiological, hereditary, behavioral factors in addition to environmental characteristics. The analysis accounted for age, sex, BMI, diabetes and socioeconomic status. At a minimum, adjustment for generally-accepted clinical risk factors for cardiovascular outcomes (e.g., ACC/AHA Pooled Cohort Equations risk factors) would be recommended in order to better address potential confounding.

Housing costs would seem to be another potential confounding variable. Within urbanized areas, housing costs negatively correlate with greenspace exposure. How are the authors thinking about housing in general, aside from percent renting and overcrowing as reflected in the Townsend Index?

The fact that physical activity did not mediate the observed relationship between greenspace exposure and CVD incidents is interesting, and coheres with other work on greenspace exposure (e.g. Maas et al., J Epi & Comm Hlth 63.12 967-973, which I see the authors have cited).

I would have liked to have seen more details on the moderation analysis results, perhaps in the form of a table or supplemental table. I couldn't follow the part of Section 3.3 that summarized the results of that analysis. Furthermore, what are some hypotheses about why individuals in certain social contexts (e.g., occupations) may demonstrate stronger/weaker associations between green space exposure and CVD events?

Living near greenspace is one mode of exposure to nature (Frumkin et al., Envir Hlth Perspectives, 2017). Residing in a green neighborhood, as characterized by these authors, represents a routine exposure to nature at an intermediate spatial scale. Other studies have indicated prolonged effects of more intense, short-term exposures to nature, such as improved natural killer cell activity (PMID: 21329564). I guess my point is that there are several ways we can think about exposure to nature, and that these may have specific and/or shared mechanisms that support improved CVD risk profiles. The authors state that the causes of the relationship between greenspace exposure and CVD are largely unexplained, and while I don't disagree with that statement, there is a growing body work in this field (as highlighted well by the Frumkin paper).

On a related note, residential location data may not represent activity spaces of individuals. The authors mention the limitation of not having record of moves that occurred between 2000 and 2014, but this would be an additional complicating aspect of characterizing individuals' exposure to nature.

The fact that physical activity did not mediate the observed relationship between greenspace exposure and CVD incidents is interesting, and coheres with other work on greenspace exposure as discussed in the above comment.

Were there steps taken to ensure the replicability of the study (e.g., data/code archiving, automated/reproducible report generation)?

Other comments:

The definition of the greenspace exposure that was given in the Abstract was very difficult to comprehend.

Likewise, the statement of the main result (7% lower relative hazard for those living in the greenest neighborhood quartile) did not specific a referent neighborhood greenspace exposure group or value.

6. PLOS authors have the option to publish the peer review history of their article (what does this mean?). If published, this will include your full peer review and any attached files.

Reviewer #1: No

Reviewer #2: Yes: Jarrod E. Dalton, PhD

---

## [Author Response · Author response to Decision Letter 0]

14 Nov 2019

Please see attached Response to Reviewers document.

---

## [Editor Report · Decision Letter 1]

2 Dec 2019

Residential neighbourhood greenspace is associated with reduced risk of cardiovascular disease: a prospective cohort study

PONE-D-19-21826R1

Dear Dr. Jones,

We are pleased to inform you that your manuscript has been judged scientifically suitable for publication and will be formally accepted for publication once it complies with all outstanding technical requirements.

With kind regards,

David Meyre

Academic Editor

PLOS ONE
---

## [Editor Report · Acceptance letter]

5 Dec 2019

PONE-D-19-21826R1 

Residential neighbourhood greenspace is associated with reduced risk of cardiovascular disease: a prospective cohort study 

Dear Dr. Jones:

I am pleased to inform you that your manuscript has been deemed suitable for publication in PLOS ONE. Congratulations! Your manuscript is now with our production department. 

With kind regards,

on behalf of

Dr David Meyre 

Academic Editor

PLOS ONE